# Impact on Public Health Nutrition Services Due to COVID-19 Pandemic in India: A Scoping Review of Primary Studies on Health and Social Security Determinants Affecting the First 1000 Days of Life

**DOI:** 10.3390/ijerph192113973

**Published:** 2022-10-27

**Authors:** Shweta Khandelwal, Mahima Mehra, Ayushi Singh

**Affiliations:** 1Public Health Foundation of India, Gurugram 122003, India; 2Triple Line Consulting Ltd., London SE1 7TY, UK; 3ASER Centre—Pratham Education Foundation, New Delhi 110029, India

**Keywords:** pregnant and nursing mothers, children under two years, 1000 days, COVID-19, India

## Abstract

Context: COVID-19 was declared ‘a global pandemic’ by the World Health Organization in March 2020. India’s lockdown, one of the harshest in the world, came with additional challenges for women. This paper aims to assess the impact of COVID-19 pandemic-related pathways on the first thousand days of life in the Integrated Child Development Scheme and the public distribution ecosystem in India. Data sources: Using Cochrane guidelines, electronic databases, namely Google Scholar and PubMed-NCBI, were searched for evidence between 1 March 2020 and 1 May 2022. A total of 73 studies were identified in initial search; 20 met the inclusion criteria and, thus, were included in the research analysis. Primary studies were conducted throughout pan-India in rural, urban, and semi-urban areas to study the impact of COVID-19 pandemic-related pathways on the first 1000 days of life. The impact of social security, food insecurity, service delivery, nutrition of pregnant and nursing mothers (P&NMs), and infant and young child feeding (IYCF) varied between geographies and within geographies. Most of the primary studies were conducted at small scale, while only three studies were pan-Indian. The majority of studies were conducted on the mental health of P&NMs and pre-natal and post-natal service delivery disruption. The paucity of the available literature highlights the need to undertake research on the impact of the COVID-19 pandemic-related pathways on 1000 days of life in India and worldwide. The best implementation practices were observed where cross-sectional programs were carried out in relation to health services and social security for P&NMs and children.

## 1. Introduction

In March 2020, COVID-19 was declared a global pandemic by the World Health Organization (WHO). By 16 May 2020, almost every country had confirmed cases of coronavirus. The unfolding of pandemic severity led to strict lockdowns and the mandating of COVID-appropriate protocols.

India imposed the world’s strictest lockdown [1].The virus, by its nature, continuously mutated and multiplied. Health systems across the world faced major shocks and were forced to prepare for the worst to come with every wave [2].

The lockdown came with dire repercussions. Joblessness decreased contact with loved ones, and online schooling and additional care responsibilities at home presented new challenges [3]. Anecdotal evidence suggests that vulnerable populations, especially women, were largely impacted as their work increased with respect to child rearing and household management [4]. Transitioning into the changing situation was difficult for all, but particularly for women, whose responsibilities expanded colossally. This is especially true for women who were pregnant or breastfeeding during the peak of the pandemic [1]. 

To begin with, pregnant and nursing mothers (P&NMs) faced considerable difficulties in accessing health care services. Transport restrictions, unavailability of health personnel, and compulsory COVID-19 testing coupled with obligations at home contributed to existing mental and physiological burdens [5,6]. The fear of the unknown, financial stress, and anxiety about contracting the disease and transferring it to the newborn were legitimate concerns for the 1.61 crore pregnant women [7] registered with India’s health authorities in March 2020. In India’s capital, Delhi, women who tested positive for COVID-19 during pregnancy were forced to undergo institutional quarantine in public health facilities with overflowing toilets and no drinking water [8]. Additionally, the pandemic saw a 20% to 68% [9] increase in domestic violence cases in India, as per data given by 18 women’s organizations. Indeed, UN Women termed domestic violence the “shadow pandemic” in early April 2020. 

Furthermore, the pandemic also impacted young children adversely. The United Nations International Children’s Emergency Fund (UNICEF) [10] estimated that the COVID-19 pandemic could result in additional two million deaths of under-five children worldwide. There was an estimated 10–50% increase in wasting seen in children under five years of age [11]. Researchers calculated that disruption of services and an increase in household food insecurity (HFI) could lead to 60% additional maternal deaths. 

Typically, P&NMs and children under two years in India are covered under the Integrated Child Development Services (ICDS) scheme and the Public Distribution System (PDS). 

Since its inception in 1975, the ICDS scheme in India is the world’s largest community-based public health system for pregnant and nursing mothers and children up to the age of six years. The ICDS scheme in India aims to improve nutritional and health outcomes and to provide a holistic environment for early childhood development. Indeed, ICDS services include pre-schooling services, supplementary nutrition, health and nutrition awareness, referral services, immunization, and health check-ups provided through a network of 1.4 million Anganwadi Centers (AWCs) [12] run by Anganwadi workers (AWWs) and Anganwadi helpers in India. In 2019, nutrition supplementation under the ICDS scheme benefitted 8.74 crore mothers and children. By 2021, it had fallen by more than 14%, owing to the COVID-19 pandemic. This was despite an increase in the allocation of funds for the ICDS. 

The PDS in India is the cornerstone of food security, poverty alleviation, and the reduction of malnutrition. It aims to provide food grains at subsidized rates to the poor through a matrix of ration cards and fair price shops (FPSs). In 2019, the PDS system fed nearly 800 million people [13] in India. During the COVID-19 pandemic, the central government announced that the Pradhan Mantri Garib Kalyan Anna Yojana [14] was providing 5 kgs of rice or wheat and 1 kg of pulses to the poor free of cost, in addition to the regular entitlement quota of food grains.

Table 1 provides an overview of central and state government provisions for health and social security during the COVID-19 pandemic.

It is in this context that we situate our research on the health, nutrition, and social security impact of the pandemic on the first 1000 days of the lives of mothers and infants under the ICDS scheme and PDS. Several researchers in India have tried to evaluate the impact of COVID-19 through a mix of primary and secondary studies. Primary studies provide a realistic picture of the situation and can be used to inform sound policy building. Hence, primary studies been selected as noteworthy sources of evidence by the research team. 

To assess the impact of COVID-19 on maternal and child nutrition, this synthesis adapts from the World Food Programme’s adaptation of maximizing the Quality of Scaling-Up Nutrition (MQSUN) nutrition framework [14] and applies it to India’s nutrition and health scenario. The COVID-19 pandemic led to disruptions in income due to unemployment and reduced work opportunities, overburdened social protection programs, decreased mobility, increased migration, competition between health emergencies and food assistance, and reduced development support. These exacerbated pandemic-related pathways, such as household food status, care for women and children (mental health), and access to health services. Advanced by existing inequities and vulnerabilities, these factors will have long-term implications for nutrition and health, such as inadequate dietary intake and disease, finally leading to all forms of malnutrition. 

The first section adapts from the framework to guide key research questions. The second section elucidates the methodology, including the theoretical framework of the key terms used throughout the paper. The third section presents the results of the research followed by a concluding discussion.

## 2. Research Question

This research aims to study the impact of the COVID-19 pandemic-related pathways on the first 1000 days of life through primary research studies carried out by researchers in rural, urban, and semi-urban areas of India. The first 1000 days of life refer to the entire duration of a woman’s pregnancy through to her child’s second birthday [15].

By doing a framework-based scoping review of the existing literature, this paper assesses the impact of COVID-19 related pathways on the following aspects:a.Access to health services for women and children including the following:
Antenatal coverage;Childbirth (delivery services for pregnant women, type of delivery, pregnancy outcomes, and complications in pregnancies);Post-natal services and immunization;Access to community health services and community health extension workers.b.Care for women and children during pandemic including the following:
Nutrition of pregnant and nursing mothers (P&NMs);Early initiation of breastfeeding, exclusive breastfeeding for the first six months of infant, continued breastfeeding for at least the first two years of life;Stress levels;Mental health, including depression, and social support.c.Household food security status (food availability, food accessibility, food choices, and food utilization);d.How did vulnerabilities of gender, social stigma, and unemployment exacerbate these indicators? What was the extent, coverage, type, and impact of social protection schemes for P&NMs and children under two years? 

## 3. Methodology

### 3.1. Theoretical Framework

Adopted from Chisvo et al.’s framework [14] this paper uses several indicators relating to health and nutrition in first 1000 days of life. Key indicators (Table 2) have been applied to the COVID-19 context in India, and its various pathways.

The meaning of *poverty* has been adapted from the Global Multidimensional Poverty Index (MPI) [16]. The MPI considers three dimensions of poverty i.e., health, education, and living standards. Acute *food insecurity* is observed when ‘a person’s life or livelihood is in danger because of the lack of food’ [17]. The WHO defines *maternal mental health* as a condition of well-being wherein a mother understands her own capacities, can adapt to the ordinary burdens of life, can work efficiently, and contribute to her society [18]. *Malnutrition* refers to lack of proper nutritional nutrient intake. This might be in deficiencies or excesses [19]. The IYCF refers to appropriate feeding practices for newborns and children under two years. Optimal IYCF practices include initial breastfeeding, exclusive breastfeeding, and complementary breastfeeding [20]. 

### 3.2. Search

This research protocol follows a Cochrane review PRISMA extension for systematic scoping reviews with a three-step process of identification, screening, and inclusion. In accordance with the Cochrane methodology, quality analysis of included studies was not performed. The desk review was conducted by MM and AS. The target group for the study was narrowed down to P&NMs and infants under two years based upon initial discussions within the research team and the corresponding author (SK). 

The search strategy was followed from title-based screening to abstract screening and finally screening of the full text of articles, as per the Cochrane collaboration. Inclusion criteria were formed at the beginning of the review, based on research questions. In Stage 1, a scoping review of published journals in two databases (Google Scholar and PubMed-NCBI) was conducted, focusing on the literature published between 1 March 2020 and 1 May 2022. Combinations of the following search key terms were applied in each database: COVID-19, India, pregnancy, service delivery, mental health, anxiety, India, nutrition, ICDS, food security, THR, IYCF, social security, and cash transfers. After the search, duplicate publications were removed. 

### 3.3. Inclusion Criteria

Primary studies carried out in India from 1 March 2020 to 1 May 2022 to study the impact of COVID-19 pandemic-related direct and indirect pathways on the first 1000 days of life (i.e., on P&NMs and infants under two years) published in peer-reviewed journals, and available online were included. Limits restricted the search to those articles with full-text published in English.

### 3.4. Synthesis

The existing literature was coded into parent codes i.e., service delivery, mental health, nutrition of mother, IYCF, food security, and social security. Further classification was carried out based on the author, year, location, study design, tool, and stakeholders involved (P&NM or infant) (Appendix A).

Finally, the retrieved literature was filtered to include only primary studies carried out on the target group, (a) covered in the ICDS ecosystem, and (b) on government provided social security services accessed through the PDS system in rural, urban, and semi-urban areas of India. Finally, a narrative synthesis approach was applied to answer the relevant research questions and compute the findings. Figure 1 provides an overview of the screening process. 

## 4. Results

### 4.1. Access to Health Services for Women and Children

#### 4.1.1. Pre-Natal and Post-Natal Services

Six studies that met the inclusion criteria were further analyzed. Out of these, five studies included details of ANC visits, institutional deliveries, referral services, and intrapartum care. One study captured details of VHND, child growth monitoring, oral rehydration solution, zinc, and THRs. Geographically, two studies were conducted in Uttar Pradesh, one in Delhi, and one in Rajasthan. Only one study met the inclusion criteria for a union territory (Puducherry). Two studies overlapped for service delivery as well as immunization services. 

The pandemic resulted in service disruption in the ICDS ecosystem on two fronts. While state and central governments across the country increased efforts towards strengthening existing platforms, frontline workers, pregnant women, and infants faced the brunt of poverty, policy implementation, and service disruptions during the pandemic. The FLWs faced challenges pertaining to transport, stress, and lack of honorarium. Additionally, P&NMs and children under two years could not access services due to the lockdown, additional care responsibilities, and fear of contracting the disease. 

Pregnant women faced considerable difficulties in accessing pre-natal and post-natal services at the community level due to lockdown. Low utilization and reduced access to services posed additional challenges for pregnant mothers. 

In Dakshinpuri, Delhi [21], a survey was conducted through a randomization of households (*n* = 4762). Out of 199 pregnant women, more women had completed 4 or more ANC visits before lockdown (97.3%) than after lockdown (84.3%). Over 100 (*n* = 103) women before lockdown received intrapartum care, while 90 (*n* = 96) women received intrapartum care after lockdown. The major challenges included poor quality of care, lack of transport to reach health providers, and long waiting times. Close to 32% of women (8 out of 25) resorted to selling jewelry, utilizing bank savings, or borrowing money to access care at private clinics. 

In Jodhpur [6] there was a 45.1% reduction (*n* = 633) in institutional deliveries. A total of 28 pregnant women had not had any antenatal visits, 10 had had single antenatal visits, and over 2.5% had to be admitted to the intensive care unit for pregnant women. A similar trend was also reported in Indore and Agra, where women (*n* = 30) missed antenatal check-ups, iron-folate tablets, and tetanus toxoid [22]. 

Cases of irregular antenatal visits were also observed in Puducherry. Out of 150 pregnant women, 41% of women did not complete the ideal number of visits and 40% developed health problems. Major challenges to health service access included economic hardships, restricted mobility, lack of information about health system changes, and psychological stress [23]. 

#### 4.1.2. Immunization Services

Studies for immunization services were found in rural areas of Bihar and Uttar Pradesh. Three studies were found meeting the inclusion criteria. All three were included in the synthesis. The most common tool for measuring immunization was in-depth interviews (IDIs) with the P&NM. 

In rural Bihar, immunization practices remained low [24]. There was a low preference for health and nutrition counselling among P&NMs according to FLWs (*n* = 30). Women preferred products, such as dry rations, over counselling services. 

In a study conducted in six community health centers and three block-level primary health centers of the District Sant Kabir Nagar in Uttar Pradesh [25], there was an overall decrease in 2.26% in the number of institutional deliveries, a 22.91% decline in antenatal services, and a 20% decline in immunization services. Nguyen et al. [5] highlighted an over 84% decline (*n* = 587) in service utilization in health and nutrition counselling in Uttar Pradesh. Reductions in growth monitoring (67%), child immunization (51%), receiving home visits (45%), and attending village health and nutrition day (VHND) (37%) were noted during the pandemic lockdown (*n* = 479).

### 4.2. Care for Women and Children

#### 4.2.1. Nutrition of P&NMs and IYCF

Four studies which met the inclusion criteria in the review focused on nutrition and IYCF. All studies evaluated the changes in nutritional intake of pregnant mothers and young children. One study was conducted online and covered pan-India. Two studies were conducted in rural and semi-urban parts of Bihar. One study was conducted in Delhi and Gurugram (Haryana). The pandemic’s impact on IYCF and nutritional intake was recorded by consumption of vegetables and meat, free rations, cooking gas, dietary score (DS), and minimum dietary diversity (MDD). 

The effects of the pandemic on child malnutrition were highlighted by UNICEF and the Population Council Institute [26] in Bihar. Indeed, 90% of households in the study (*n* = 794) reduced their consumption of vegetables and meat. Pregnant women consumed three-fifths of the recommended food groups (vegetables, non-vegetarian foods, fruits, pulses, cereals, and dairy products), but reduced consumption of dairy and fresh foods. Only 29% of households with under-6 children received cash for mid-day meals. Researchers in Delhi and Gurugram [2] interviewed migrant women accessing government welfare services (*n* = 19). The closure of AWCs impacted women and children negatively. The nutrition of children and mothers was impacted due to the disruption of mid-day meals in schools, the lack of availability of cooking gas, and the rising price of vegetables. This pushed migrant workers, who were already jobless, into additional cycles of poverty. 

In rural Bihar [27] the mean DS of mothers with children aged from 3 to 36 months decreased from 5.65 to 5.20 from the first phase of the survey (January–March 2020) to the second phase (October–November 2020). The MDD declined by 11% (*n* = 1148). Furthermore, 92% of households obtained free rations, and over 81% benefitted from cash transfers. The impact on infant health was positive [27]. Amongst children aged 25–36 months, dietary scores increased for both boys and girls by over 11%. Access of households to security schemes was high. Indeed, 92% of households obtained free rations, while 81% received cash transfers [27].

Misinformation and fear related to breastfeeding and infant feeding were rampant. In an online survey [28] using snowball sampling, over 26% (*n* = 472 women of reproductive age) agreed that COVID-19 can be passed onto the baby through breastfeeding. Furthermore, 24% of health workers (*n* = 322) believed that breastfeeding can transmit COVID-19. Such assumptions have dire consequences for exclusive and supplementary breastfeeding practices at community levels. 

#### 4.2.2. Mental Health including Depression and Social Support

Eleven studies found that COVID-19 had an impact on the mental health of P&NMs in India. Out of these, six studies met the inclusion criteria and were included for analysis. Geographically, two studies were conducted, with one across South India and one in Tamil Nadu. In North India, two were conducted in Uttarakhand, and one was conducted in Delhi. 

Various tools were used by researchers, namely the Generalized Anxiety Disorder-7 (GAD-7), impact of Events Scale-Revised (IES-R) and the Depression, Anxiety, and Stress Scale (DASS21), to measure the impact of the pandemic related disruptions on P&NMs. Most of the studies involved assessing stress, anxiety, and depression as consequences of direct and indirect pathways. 

In focus group discussions and IDIs conducted across pan-India [29] with 14 pregnant and 11 postpartum women chosen through purposeful sampling, both social and emotional factors were found to contribute to the mental health of P&NMs. 

Two primary studies conducted in South India make for a strong inquiry. Prevalent concerns included hospital visits, the safety of the child, and anxiety about COVID-19. Firstly, in Tamil Nadu, Anandhi et al. [30] used DASS21 to compare levels of psychological impact between COVID positive and negative perinatal mothers (*n* = 300). Depression (67.7%), stress (53%), and anxiety (19%) were rampant in both groups. Four mothers in the control group and one mother in case group recorded severe depression. All mothers recorded some form of anxiety in the case-control and control group. 

Secondly, in an online survey in South India, pregnant women raised concerns to their obstetricians (*n* = 118) regarding contracting the COVID-19 infection (39.83%), the safety of their infant (52.14%), concerns related to hospital visits (72.65%), methods of protection against COVID-19 (60.17%), and anxieties related to social media messages (40.68%) [31].

In Northern India, studies were carried out in New Delhi, Dehradun, Bhubaneshwar, Dehradun, Haridwar, and Nainital. Two cross-sectional surveys conducted in three districts of Uttarakhand [32] and a hospital-based study conducted in New Delhi, Dehradun, and Bhuwaneshwar [33] concluded a minimal psychological impact and anxiety levels in pregnant women during COVID-19. Tikka et al. (31) reported that 11.1% of pregnant women (*n* = 620) were rated to have anxiety disorder scores of less than 10 on the Generalized Anxiety Disorder-7 (GAD-7) scale. More than half of the sampled pregnant women (64.2%) showed no anxiety, while 24.7%, 8.5%, and 2.6% had mild, moderate, and severe anxiety, respectively. The most common reasons of fears were related to the risk of infection to the fetus (40.6%), the risk of contracting COVID herself (37.7%), and becoming infected during labor/delivery (36.6%).

Similar findings were reported by Jelly et al. [32], where most women (73.6%, *n* = 333) who showed a nominal psychological impact on an IES-R score between 0–23. 18.3% had a mild psychological impact (IES-R score of 24–32). Of these, 3% had moderate impact, while 5.1% had severe psychological impact with IES-R score of 33–36 and ≥37, respectively. Additionally, similar scores were observed on the GAD-7 scale. Awareness regarding COVID (32) employment status, presence of financial loss, semi-urban habitat, complications in present or past pregnancies, COVID testing status, lower perceived social support, and greater COVID risk perception were positively associated with anxiety in pregnancies during pandemic [33]. 

Nilima et al. [34] conducted a pan-India cross-sectional online survey (*n* = 1316). In it, 50% of mothers reported sleep cycle disruption, and over 90% reported worries related to their child’s health and contracting the disease. 

### 4.3. Household Food Security Outcomes

Researchers have used several tools to map the severity of food insecurity in a household and its consequences on child mortality during the pandemic years. These include the Household Food Insecurity Access Scale (HFIAS) for HFI measurement and the Lives Saved Tool (LiST) for modeling exercises. Three studies were found through the methodology. All were included in the analysis. Most of the studies were either pan-Indian or included India in their analysis. Only one study was focused on Uttar Pradesh. The study at Azim Premji University [35] contained information for both food security and social security. 

In India, a longitudinal survey [36] was conducted with mothers who have children younger than 2 years old (*n* = 256) in 26 blocks of 2 districts in Uttar Pradesh in December 2019 (in-person survey) and in August 2020 (by phone). Furthermore, HFIAS was used to measure HFI, which increased drastically from 21% (December 2019) to 80% (August 2020). Maternal recall of all foods and liquids consumed by the child in the previous 24 h was collected based on the categorization by WHO guidelines. Children living in households that reported an increase in HFI, were less likely to consume a diverse diet (18%) compared to children living inconsistently food-secure households (28%), where MDD was defined as consumption of four or more food groups.

Finally, researchers at Azim Premji University [35] conducted phone interviews of over 5000 respondents in 12 states of India. Although the number of pregnant and lactating women in this sample was not specified, the authors noted that 90% of households had faced a reduction in food intake due to the lockdown. Households in urban areas were worse-off than rural households. Indeed, 28% of households in urban areas highlighted that food intake had not improved even six months after the lockdown, while in rural households, food intake improved by 13%. 

### 4.4. Social Protection Schemes for Women and Children

The COVID-19 pandemic disrupted social security services to a great extent. Public policies were put in place to protect the vulnerable. However, on-ground realities presented a mixed picture. Six primary studies were included in the analysis. The authors concluded that the implementation of social security schemes varied by geography. 

Kesar et al. [35] carried out a phone survey across 12 states of India of families eligible for cash transfers under Pradhan Mantri Jan-Dhan Yojana (PMJDY). Over 50% of households had women as the owners of the Jan-Dhan account. The authors concluded that 40–60% (*n* = 5000) did not receive any cash transfers. Nearly 80% accessed the PDS system. However, PDS support was provided only to permanent residents of the State. 

In Delhi and Gurugram [2], the majority of the 19 migrant women interviewed highlighted that they did not have a ration card. Although in some cases the ration card requirement was relaxed by PDS work, many did not get any support due to documentation concerns. Few participants received support from local NGOs in the form of rations and groceries. In a cross-sectional survey of Bihar and U.P. [37] only 25–33% of eligible households (*n* = 1694 households with pregnant women and <5 years children) received take-home rations due to the closure of Anganwadi centers. There was a drop in the number of free meals provided to pregnant women under JSSK. 

In a study in Dharavi, Mumbai (*n* = 450 families), 84.8% households purchased grains from government-subsidized stores under PDS, 58.8% received them for free, 44% received cooked food/rations from ICDS or their employers, and 34% received food from NGOs [38]. In U.P. [36] 71.2% of women received THR and 22.6% received dry rations (*n* = 569 mothers with children < 2 years). However, no mother in the survey received hot cooked meals during lockdown. 

In April 2020, a rapid assessment by the Karnataka government evaluated that only two-thirds (75%) of malnourished P&NMs and 13.75% of adolescent girls received rations [39] in the first two months of the lockdown. The mandatory egg and milk were not provided for children aged 0–6 years in over 80 villages and localities. Delivery of pre-schooling activities was suspended by the Ministry of Home Affairs, disrupting the supply of hot cooked meals and feeding services.

There was an urgent need to maximize the coverage and reduce the delivery costs in India. In Chhattisgarh [40], over 24,000 people across 19 slums could not access social security services due to the absence of below poverty line (BPL) cards. There was an excessive stigma related to patients who returned from institutional COVID-19 quarantine in the early stages of the lockdown. 

## 5. Discussion

This paper provides a scoping review of the pan-Indian impact of COVID-19 pandemic-related pathways on the first 1000 days of life, by integrating concepts from the ICDS and PDS ecosystem within India, as well as from the WHO, UNICEF, and OPHI, globally. With the synthesis, we draw several linkages between matrixes of social security, food insecurity, nutrition of the P&NMs, IYCF, and service delivery.

### 5.1. Synergies between the Socio-Economic Impact of the COVID-19 Pandemic in LMICs

Evidence of the impact of the COVID-19 pandemic on P&NMs and children under two years in low- and low middle-income countries (LMICs) is limited. This paper only includes insights from India; however, during the data collection stage, synergies between different LMICs were found. The risk of stillbirth, low birth weight, and small for gestational age was experienced by most LMICs across the globe [41]. Previous researchers have highlighted the impact of COVID-19 on reduced breastfeeding practices [42] and childhood wasting [43] in over 94 LMICs. 

It can be inferred from the results of review that the imposition of the lockdown and other distancing protocols exposed the inequitable socio-economic fabric of the country. Vulnerable groups, such as women and children in government-provided social security-dependent households, were particularly hard hit due to the pandemic related pathways. Present research concludes that these reductions in service utilization were due to lockdown related measures in India. These findings are consistent with Ahmed et al. [44] and Roy et al. [45] who synthesized evidence from other LMICs on the impact of the pandemic on maternal and child health related concerns. Ahmed et al. [44] explored contextual factors of health of P&NMs and infants in Bangladesh, Nigeria, and South Africa. They highlighted that the decline of services was due to lockdown related measures and a lack of provisions for FLWs. Roy et al. [45] synthesized 170 articles from Medline/PubMed and Google Scholar in 2020. Researchers highlighted the reduction in service utilization in LMICs including Uganda and India. 

### 5.2. Varied Impacts of Social and Food Security Policies

In India, while social security policies through the PDS for the poor and vulnerable were put in place by the central and state governments, the implementation and impact of the policies varied between states and, sometimes, within states. Families of P&NMs and infants faced challenges in accessing THR due to a lack of ration-cards and migration. In April 2021, the World Food Programme [46] reported that take-home rations (THR) were being delivered through doorsteps monthly/once every two months in Punjab, Haryana, Uttar Pradesh, Manipur, Mizoram, West Bengal, Bihar, Goa, and Chandigarh. In Nagaland, THRs were being delivered daily, while FLWs in Madhya Pradesh, Tamil Nadu and Assam organized it fortnightly or weekly. Over 16 states and UTs included dry rations in their THR baskets. Bihar provided cash entitlements and dry rations. 

In Delhi, there was a reduction in ANC visits after the lockdown [21]. Factors, such as rising prices of cooking fuel and vegetables, affected nutritional choices and dietary diversity. Older children could not access mid-day meals. Mental health concerns were raised in cross-cutting studies [2,33] due to factors of unemployment, service delivery, and lockdown. Uttar Pradesh and Bihar were also significant sites of investigation. In Uttar Pradesh, studies on P&NMs underlined concerns of antenatal and post-natal service delivery, immunization services, and food insecurity. In parts of U.P., FLWs scaled up outreach activities, such as immunization and VHND. Women received free food during hospital stays. However, at the same time, HFI increased by 80% (*n* = 587). In Bihar, even though immunization practices were operational, the attendance by P&NMs and infants remained low. In a survey of 1148 women and children, households succeeding in obtaining free rations (92%), and 81% benefitted from cash transfer. In the same survey, it was concluded that the MDD of mothers decreased. Hence, the impact of policies was unequivocal even within the same population. 

### 5.3. Role of Frontline Workers

Frontline workers were the cornerstone of the global fight against the COVID-19 pandemic. Bennett et al., 2020 documented the experiences of FLWs over Twitter during the first wave of the pandemic in the UK. The FLWs experienced mental trauma during care of COVID patients. Many FLWs contracted the virus and faced death. Our findings present similar results. Even when FLWs were not included in the analysis, several other studies on service delivery included FLWs. Despite the best efforts of the government, FLWs encountered various challenges while delivering essential health and nutrition services. Some common concerns included less family support, stress, and fear of contracting the virus [47]. In Odisha, logistical challenges faced by FLWs involved issues pertaining to transportation and inadequate personal protective equipment [47]. Community-level challenges, such as stigma and resistance in some communities, leading to hindrances in the delivery of MCH, services were also reported [47]. Researchers in rural Bihar [24] concluded that several factors contributed to the performance and support of frontline workers. In in-depth interviews with FLWs (*n* = 30), family income, familial and social pressures, workload management, and the need for an honorarium were highlighted as factors influencing the performance of health workers.

### 5.4. Inconsistencies in Data

The paucity of studies performed is rampant in the analysis presented. As primary studies carried out in the pandemic included a mix of phone surveys, online surveys, and on-ground data collection, the demographic profile of mothers (rural, urban, and semi-urban) could not always be deciphered. Some single center case studies have been added in the synthesis as they highlight health systems research and the impact of the COVID-19 pandemic on the overall service delivery and information systems. 

### 5.5. A New Opportunity for Researchers

From a research perspective, the pandemic allowed researchers to observe changes in the methods of data collection and the implementation of research. Methods of data collection shifted from on-ground to online in lieu of adaptations to the present challenges. In our research, we assessed that while some researchers chose to compare data sets before and during the pandemic [5], others used technology i.e., phone surveys and online questionnaires [31,48], to reach impacted families in the pandemic in India. Methods of sampling also changed based upon who had on-ground presence initially and finally, leading to partnerships and collaborations where possible. Lobe et al. (2020) also presented familiar findings on the rise of internet-based data collection during the COVID-19 pandemic. Computer based research methods offer flexibility of time and distance. Even though they comes with ethical challenges, these methods have the potential to document experiences in far-off communities as never before, especially after the pandemic years. If *‘leaving no one behind’* is a global goal for the decade, it has been made possible by public health researchers now more than ever.

## 6. Conclusions

Though the evidence presented on the impact of the COVID-19 pandemic on the first 1000 days of life is limited, the findings of the primary studies open up avenues of further research on P&NMs and children under two years. The paucity of the studies available underlines how P&NMs and children were missed in research priorities and policy building. The paper underscores the critical role of government-provided health services and social security ecosystem in protecting the vulnerable from the aftermath of the pandemic years in India. There is varied impact between and within geographies in terms of the service delivery of health and nutrition under ICDS system. 

In terms of implementation of existing programs in India, results were seen best where cross-sectional programs for health and nutrition were rolled out for P&NMs and infants. Improvements in maternal and child health outcomes require coordinated action on service delivery, food security, IYCF practices, and social security. All of these factors together shape maternal health aspects. While efforts were made to protect the vulnerable mothers and infants from facing the worst consequences of the pandemic, we must not deny the realities that many were pushed into additional poverty, acute hunger, and food insecurity. The authors highlight the need for policies that are mindful of these realities and actively aim to overcome them. If efforts are not made to improve the nutrition, health, and social security of infants and mothers, we might be looking at another pandemic of successive poverty traps and declining global health outcomes.

## 7. Limitations

Due to the nature of the research question, we only included primary studies conducted during the COVID-19 pandemic to map the impact of the pandemic on P&NMs and children under two years. Several secondary studies also provide important insights into the maternal-child health scenario in India and elsewhere. These studies informed our perspectives about the landscape of the research and are equally relevant to public health and nutrition research in LMICs, especially in India. Limitations also include the small study samples in the evidence presented. This is due to the difficulties in accessing the target group due to COVID-19 related disruptions. Finally, the authors acknowledge the variation between the study designs of the evidence gathered. This is due to the paucity of data available. 

## Figures and Tables

**Figure 1 ijerph-19-13973-f001:**
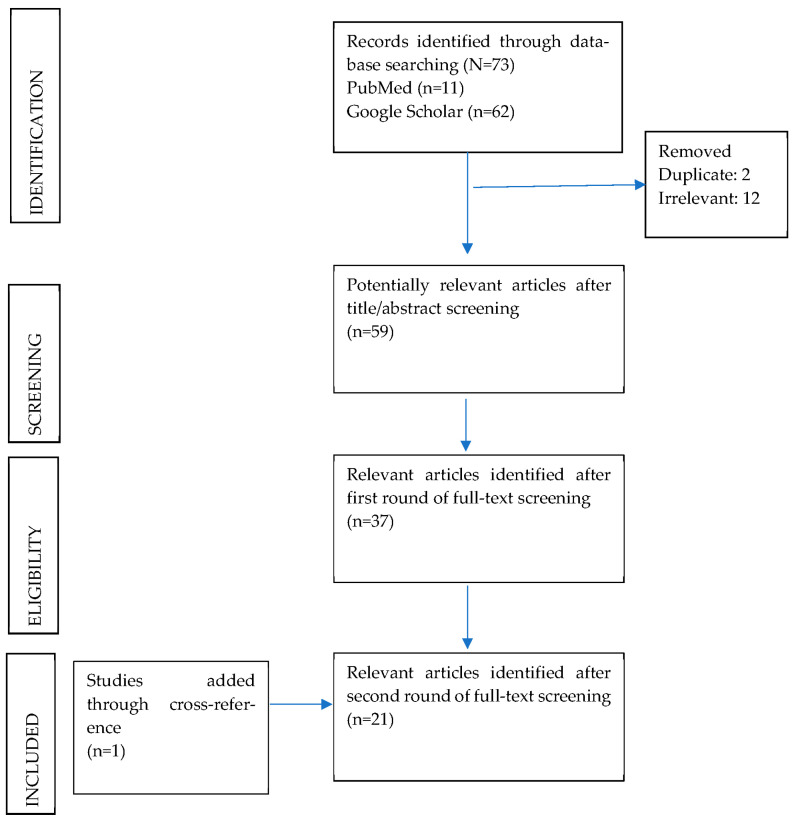
PRISMA diagram of scoping review.

**Table 1 ijerph-19-13973-t001:** Overview of health and social security provisions under ICDS and PDS in India.

Health and Social Security Provision	Government Body	Efforts during COVID-19 Pandemic
Integrated Child Development Scheme	Ministry of Women and Child Development, Government of India	On 30 March 2020, the Ministry instructed AWWs to provide a single instalment of food grains or meals to mothers, infants, and adolescent girls to the extent possible, and to continue door-to-door service delivery if needed.
	Uttarakhand State Government	Promised life insurance worth 4 lakh INR to 68,000 health workers under ICDS (Hindustan Times, 2020) including Anganwadi workers, Anganwadi assistants, and employees of the Garhwal Mandal Vikas Nigam and Kumaon Mandal Vikas Nigam.
	Madhya Pradesh Government	Promised health insurance coverage of INR50 lakhs to frontline workers (FLWs).
Public Distribution System	Central Government	Cash transfers of INR500 were initiated by the central government under Pradhan Mantri Jan Dhan Yojana (PMJDY) (Sridhar, 2021). Additionally, all families under the public distribution system were allocated a free 1 kg of pulses for three months (Economic Times, 2020). Pradhan Mantri Garib Kalyan Anna Yojana (PMGKAY) for NFSA beneficiaries and Atma Nirbhar Bharat Package (ANB) for Migrants/Stranded Migrants.
	Bihar state Government	Community kitchens and relief camps were initiated in government schools to provide food and shelter.
	Delhi State Government	Temporary ration cards were developed to provide e-coupons, wheat, and rice without charge to all poor communities.
	Uttar Pradesh Government	Started dry ration distribution only in October 2020. While the government promised Rs 50 L relief to health workers, in January 2020 only 16% (12) out of the total 72 Anganwadi workers who died on COVID duty had received insurance support from the government (The Print, 2022).

**Table 2 ijerph-19-13973-t002:** Summary of key indicators.

Domains	Target Group	Key Indicators
Pre-natal and post-natal service delivery	P&NMs	TT1/TT2/Booster, ANC check-ups (at least 4), institutional deliveries, type of delivery, place of delivery, village health and nutrition days (VHND), cash transfers under the Pradhan Mantri Matru Vandana Yojana Scheme (PMMVY), Janani Suraksha Yojana (JSY)
Immunization services for children under two years	Children under two years of age	Six vaccine preventable diseases covered under ICDS, namely poliomyelitis, diphtheria, pertussis, tetanus, tuberculosis, and measles
Nutrition of P&NMs	P&NMs	IFA (≥100 days), deworming, calcium (360 tablets), THR, counseling for the diet of mother and child
IYCF practices	P&NMs, Children under two years of age	Early initiation of breastfeeding, exclusive breastfeeding for the first six months, continued breastfeeding for at least first two years of life, diet diversity, adequate quantity and quality of food, child growth monitoring, oral rehydration solution
Food insecurity of households	Household of a P&NM with children under two years of age	Food insecurity, relation of HFI to diet of mother and child, PDS system, social protection under Pradhan Mantri Garib Kalyan Yojana
Mental health of pregnant mother	P&NMs	Direct pathways, namely stress, anxiety, panic, and depression due to disruption of antenatal and post-natal services, immunization challenges for the infant, fear of testing positive for COVID-19
		Indirect pathways, namely stress, anxiety, panic, and depression due to domestic abuse, food insecurity, loss of employment, social interaction, and perceived social support

## Data Availability

Not applicable.

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
