# Peer review of "Impact on Public Health Nutrition Services Due to COVID-19 Pandemic in India: A Scoping Review of Primary Studies on Health and Social Security Determinants Affecting the First 1000 Days of Life"

_ijerph, 2022, doi:10.3390/ijerph192113973_

Round 1

Reviewer 1 Report

This study includes important information about how the COVID-19 pandemic impacted aspects of health in India. It's crucial to disseminate these findings. However, the paper is difficult to follow and it would be helpful to restructure the paper to make the points more concise and direct. A specific aim and hypothesis are needed to direct the manuscript, there seems to be too many aspects attempting to be covered and it makes it difficult for the reader to follow the paper. Although the paper states the focus is on the first 1,000 days of life for mothers and and children, the studies included in this review don't seem to necessarily follow this narrative. 

Abstract:

line 12: add a comma after "world"

line 14: spell out ICDS and PDS the first time the acronyms are used in case readers are unfamiliar 

lines 16-17: suggest rephrasing "73 studies were identified in initial search, while 20 were included in the research analysis meeting the inclusion criteria" to something like "73 studies were identified in initial search and 20 met the the inclusion criteria, thus were included in the research analysis."

line 20: please spell out P&NM and IYCF

line 21: suggest "most of the studies" instead of maximum 

line 24: suggest starting sentence with "The"

Introduction:

line 30: suggest removing the quotes around global pandemic

line 34: was this truly the harshest lockdown across the world? perhaps using "one of the strictest lockdowns" since other areas had strict lockdown rules and this can be subjective--can you please also include a description of the details of the lockdown?

lines 35 & 36: add citation to support this

line 37: add citation

line 39-41: add citation

lines 41-42: add citation

lines 42-43: add citation

lines 44-45: add citation

lines 45-47

**overall all these statements need supporting evidence

line 56: add comma before "as well"

line 56: spell out UNICEF

lines 58-59: suggest rephrasing to something like "There was an estimated 10-50% increase in wasting seen in children under the age of five." and then cite the Lancet 

line 59: spell out HFI

line 60: remove "of"

lines 71-73: add citation 

line 88: remove "on-ground"--unclear what this means

lines 90-91: suggest using research team instead of authors and remove author initials

Table 1: suggest writing out acronyms in table and changing the spacing between words to be more close together--perhaps add a column to show if these were effective or positive assistances or no changes

line 94: what is meant by the "Indian context?" Is this trying to state applying the nutrition framework to India? or a specific model?

line 98: add comma before "such as"

line 105: remove "we proceed as follows"

line 111: primary research studies?

Table 2: same comment about changing the spacing between words

lines 152-153: Recommend making this more general: "The research team conducted the initial search..."

line 172: can remove MS Excel

line 175: remove "This provided a purposeful summary of the findings."

line 241: suggest using "four studies met the inclusion...." instead of meeting

line 249: please elaborate what compromised in this context means

line 250: what food groups did they meet? what are these recommendations?

lines 256-270: add citations

line 273: Suggest using "Eleven studies found that COVID-19 had an impact on...."

line 274: suggest "Of these, six studies met the inclusion criteria and were included for analysis"

line 275: remove extra space between India + and

line 282: focus group discussions?

line 285: unclear what this sentence is stating--suggest rephrasing 

lines 327-335: please make the spacing after periods consistent 

line 361: is "slum" the appropriate term to use?

lines 385-387: suggest rephrasing, sentence is unclear

lines 455-456: suggest rephrasing--the pandemic was an extremely difficult time and it comes off insensitive to say it was a novel time for research

line 491: capitalize COVID to be consistent & remove extra spacing between on and P&NM

Author Response

Reviewer 1

  1. Overall: This study includes important information about how the COVID-19 pandemic impacted aspects of health in India. It's crucial to disseminate these findings. However, the paper is difficult to follow and it would be helpful to restructure the paper to make the points more concise and direct. A specific aim and hypothesis are needed to direct the manuscript, there seems to be too many aspects attempting to be covered and it makes it difficult for the reader to follow the paper. Although the paper states the focus is on the first 1,000 days of life for mothers and children, the studies included in this review don't seem to necessarily follow this narrative.

Response: We thank the reviewer. We have framed a research question and specific layers we have attempted to review from lines 110-133 in the revised draft. The results also have been provided under the 4 identified pathway themes to make things clearer. The papers identified by our search had 1000 days as one of the inclusion criteria. For some papers which go beyond 1000 days, we have limited our analysis to the results provided uptil 1000 days. We hope this will assuage the reviewer’s concern.

  1. Abstract:
  • line 12: add a comma after "world"
  • line 14: spell out ICDS and PDS the first time the acronyms are used in case readers are unfamiliar
  • lines 16-17: suggest rephrasing "73 studies were identified in initial search, while 20 were included in the research analysis meeting the inclusion criteria" to something like "73 studies were identified in initial search and 20 met the the inclusion criteria, thus were included in the research analysis."
  • line 20: please spell out P&NM and IYCF
  • line 21: suggest "most of the studies" instead of maximum
  • line 24: suggest starting sentence with "The"

Response: All the above suggestions have been addressed in track changes in the manuscript.

  1. Introduction:
  • line 30: suggest removing the quotes around global pandemic
  • line 34: was this truly the harshest lockdown across the world? perhaps using "one of the strictest lockdowns" since other areas had strict lockdown rules and this can be subjective--can you please also include a description of the details of the lockdown?
  • lines 35 & 36: add citation to support this
  • line 37: add citation
  • line 39-41: add citation
  • lines 41-42: add citation
  • lines 42-43: add citation
  • lines 44-45: add citation
  • lines 45-47 **overall all these statements need supporting evidence
  • line 56: add comma before "as well" line 56: spell out UNICEF
  • lines 58-59: suggest rephrasing to something like "There was an estimated 10-50% increase in wasting seen in children under the age of five." and then cite the Lancet line 59: spell out HFI
  • line 60: remove "of" lines 71-73: add citation
  • line 88: remove "on-ground"--unclear what this means
  • lines 90-91: suggest using research team instead of authors and remove author initials

Response: All the above suggestions have been addressed in track changes in the manuscript.

  1. Others:

Table 1:

  • suggest writing out acronyms in table and changing the spacing between words to be more close together--perhaps add a column to show if these were effective or positive assistances or no changes
  • line 94: what is meant by the "Indian context?" Is this trying to state applying the nutrition framework to India? or a specific model?
  • line 98: add comma before "such as"
  • line 105: remove "we proceed as follows"
  • line 111: primary research studies?

Table 2: same comment about changing the spacing between words

  • lines 152-153: Recommend making this more general: "The research team conducted the initial search..." line 172: can remove MS Excel
  • line 175: remove "This provided a purposeful summary of the findings."
  • line 241: suggest using "four studies met the inclusion...." instead of meeting
  • line 249: please elaborate what compromised in this context means
  • line 250: what food groups did they meet? what are these recommendations?
  • lines 256-270: add citations
  • line 273: Suggest using "Eleven studies found that COVID-19 had an impact on...."
  • line 274: suggest "Of these, six studies met the inclusion criteria and were included for analysis"
  • line 275: remove extra space between India + and
  • line 282: focus group discussions?
  • line 285: unclear what this sentence is stating--suggest rephrasing
  • lines 327-335: please make the spacing after periods consistent
  • line 361: is "slum" the appropriate term to use?
  • lines 385-387: suggest rephrasing, sentence is unclear
  • lines 455-456: suggest rephrasing--the pandemic was an extremely difficult time and it comes off insensitive to say it was a novel time for research
  • line 491: capitalize COVID to be consistent & remove extra spacing between on and P&NM

Response: All the above suggestions have been addressed in track changes in the manuscript.

Reviewer 2 Report

 This was an interesting review that assesses the impact of COVID-19 pandemic-related pathways on the first thousand days of life in the ICDS and PDS ecosystem in India. The manuscript is certainly unique in its own right, however, there are some major concerns worth raising.

My primary concern with this review involves the Methods section.

Firstly, I suggest that the authors use a The PRISMA extension for scoping reviews (https://www.prisma-statement.org/Extensions/ScopingReviews) as it will add merit to the methodology.

Secondly, the use of additional electronic academic databases would likely have added to the study in both complexity and sample size. If the scope of this study were expanded to use additional databases, more sources might have been identified and explored. For example, the following electronic databases could have been used for a more thorough and inclusive search;

“ArticleFirst; Biomed Central; BioOne; BIOSIS; CINAHL; EBSCOHost; JSTOR; ProQuest; PubMed; SAGE Reference Online; Scopus; ScienceDirect; SpringerLink; Taylor & Francis; and Wiley Online.” These databases would have likely added to the overall literature search in their academic rigor, aim, and biomedical scope. While Google Scholar and PubMed are good database, the use of more databases would have added to the study sample size.

Overall, this paper does not merit publication in its current form.

Author Response

This was an interesting review that assesses the impact of COVID-19 pandemic-related pathways on the first thousand days of life in the ICDS and PDS ecosystem in India. The manuscript is certainly unique in its own right, however, there are some major concerns worth raising. My primary concern with this review involves the Methods section.

Firstly, I suggest that the authors use a The PRISMA extension for scoping reviews (https://www.prismastatement.org/Extensions/ScopingReviews) as it will add merit to the methodology.

Response: We have added this under Section 3.2. Edits are made in track changes in the manuscript.

Secondly, the use of additional electronic academic databases would likely have added to the study in both complexity and sample size. If the scope of this study were expanded to use additional databases, more sources might have been identified and explored. For example, the following electronic databases could have been used for a more thorough and inclusive search.

“ArticleFirst; Biomed Central; BioOne; BIOSIS; CINAHL; EBSCOHost; JSTOR; ProQuest; PubMed; SAGE Reference Online; Scopus; ScienceDirect; SpringerLink; Taylor & Francis; and Wiley Online.” These databases would have likely added to the overall literature search in their academic rigor, aim, and biomedical scope. While Google Scholar and PubMed are good database, the use of more databases would have added to the study sample size.

Overall, this paper does not merit publication in its current form.

Response: We are very sorry that the reviewer feels this way. However, this paper is a scoping review attempted to provide an overarching perspective on the impact of COVID-19 on pregnant and nursing mothers and children (under-2 years) in India. It also attempts to understand the nature and type of research attempted during the lockdown years by researchers. It does not claim to be exhaustive. We regard a smaller sample size as evidence and the same has been illustrated in the limitations of the paper. We sincerely hope this will be acceptable to our esteemed reviewer.

Round 2

Reviewer 1 Report

The authors have incorporated comments from the review suggestion--thank you!

Reviewer 2 Report

thank you for addressing and responding to these recommendations.